# Evaluating the impact on health outcomes of an event that resulted in a delay in contact tracing of COVID-19 cases in England, September 2020: an observational study

Lucy Findlater ,[1] Livia Pierotti,[2] Charlie Turner,[3] Adrian Wensley,[4] Cong Chen ,[3] Shaun Seaman ,[5] Pantelis Samartsidis ,[5] Andre Charlett ,[6] Charlotte Anderson,[6] Gareth Hughes ,[4] Matthew Hickman ,[2] Obaghe Edeghere ,[7] Isabel Oliver [1]

For numbered affiliations see end of article.

**Correspondence to**
Livia Pierotti;
livia.pierotti@bristol.ac.uk

## ABSTRACT

**Objective** In September 2020, 15 861 SARS-CoV-2 case records failed to upload from the Second Generation Surveillance System (SGSS) to the Contact Tracing Advisory Service (CTAS) tool, delaying the contact tracing of these cases. This study used CTAS data to determine the impact of this delay on population health outcomes: transmission events, hospitalisations and mortality. Previously, a modelling study suggested a substantial impact.

**Design** Observational study.

**Setting** England.

**Population** Individuals testing positive for SARS-CoV-2 and their reported contacts.

**Main outcome measures** Secondary attack rates (SARs), hospitalisations and deaths among primary and secondary contacts were calculated, compared with all other concurrent, unaffected cases. Affected SGSS records were matched to CTAS records. Successive contacts and cases were identified and matched to hospital episode and mortality outcomes.

**Results** Initiation of contact tracing was delayed by 3 days on average in the primary cases in the delay group (6 days) compared with the control group (3 days). This was associated with lower completion of contact tracing: 80% (95% CI: 79% to 81%) in delay group and 83% (95% CI: 83% to 84%) in control group. There was some evidence to suggest increased transmission to non-household contacts among those affected by the delay. The SAR for non-household contacts was higher among secondary contacts in the delay group than the control group (delay group: 7.9%, 95% CI: 6.5% to 9.2%; control group: 5.9%, 95% CI: 5.3% to 6.6%). There did not appear to be a significant difference between the delay and control groups in the odds of hospitalisation (crude OR: 1.1 (95% CI: 0.9 to 1.2)) or death (crude OR: 0.7 (95% CI: 0.1 to 4.0)) among secondary contacts.

**Conclusions** Our analysis suggests that the delay in contact tracing had a limited impact on population health outcomes; however, contact tracing was not completed for all individuals, so some transmission events might not be captured.

## STRENGTHS AND LIMITATIONS OF THIS STUDY

⇒ Shows empirical data on the health impact of an event leading to a delay in contact tracing so can test hypotheses generated by models of the potential impact of a delay in contact tracing.

⇒ Estimates the extent of further transmission and odds of increased hospitalisation or mortality in up to the third generation of cases affected by the event.

⇒ The event acts as a natural experiment to describe the possible impact of contact tracing, comparing a group affected by chance by delayed contact tracing to a control group who experienced no delay.

⇒ Contact tracing was not completed for all individuals, so the study might not capture all affected contacts or transmissions.

## INTRODUCTION

As of October 2023, there have been over 22 million cases and 200 000 deaths from COVID-19 reported in the UK.[1] Contact tracing has been a central part of the public health response to SARS-CoV-2 and involves identifying contacts of people who have tested positive and advising them to self-isolate to reduce onward transmission.[2] On 28 May 2020, a national contact tracing system, National Health Service (NHS) Test and Trace, was launched. All PCR-positive cases in England were identified and contacted by phone, digital tools and through partnerships with local authorities.[3 4] Between 25 September and 2 October 2020, some data files containing test results from community testing sites failed to import from the laboratory surveillance system (Second Generation Surveillance System (SGSS)) to the contact tracing tool (Contact Tracing and Advisory

Service (CTAS)). The files contained 15 861 new positive cases.[5] At that time, on 28 September 2020, there was a 7-day average of around 8200 reported cases per day in England with significant regional variation.[1]

The event did not affect the notification of test results to patients or affect the results of any specific testing sites, geographical areas or population groups. However, the delay in the upload of case records to CTAS resulted in a delay in the contact tracing of these cases, and therefore a potential delay in instructing their contacts to self-isolate, which could have resulted in further transmission of SARS-CoV-2, and potentially more COVID-19-related hospitalisations and deaths. We expect that health outcomes of the initial group of cases (primary cases) whose records failed to upload to CTAS would not have been affected, as they received their test results and advice to isolate in the usual timeframe, and transmission events from primary cases to their contacts (primary contacts who may become secondary cases) would have most likely occurred prior to the isolation of the case. However, these primary contacts may not have been aware that they were a close contact of a case or advised to isolate in a timely manner, which may have increased the risk of transmission to their contacts (secondary contacts who may become tertiary cases). The outcomes of secondary contacts are therefore the most likely to have been impacted by the delay in contact tracing. In addition, there could potentially be delays in secondary cases seeking healthcare support, increasing their risk of hospitalisation or mortality.

A published preliminary model suggested that this event may have been associated with an increase of up to a third additional infections and 30–40% additional deaths.[6]

In this study, we assess directly the health impact of the event, in terms of transmissions, hospitalisations and deaths, using data on affected and concurrent cases and their contacts.

### Aims and objectives

The aims of this study are to:

► Describe the nature of any contact tracing delay experienced by cases affected by the event.
► Assess the extent of further transmission, by calculating attack rates from primary cases to their contacts and from secondary cases to their contacts.
► Assess the impact of the delay on health outcomes of primary and secondary contacts, in terms of hospitalisations and deaths.

### METHODS
### Data sources

The list of SGSS records affected by the event was provided by the UK Health Security Agency (UKHSA). SGSS contains demographic and diagnostic information from laboratory test reports for patients in England who test positive for SARS-CoV-2.[7] SGSS records were matched to COVID-19 testing and contact tracing records from the CTAS database to validate the cases affected by the

event. CTAS records represent SARS-CoV-2 case episodes, including information on the movements of cases in their infectious period, their contacts and demographic and clinical characteristics (online supplemental figure 1).[8] Matching was conducted in repeated rounds based on combinations of the following identifiers: SGSS unique identifier; NHS number; forename; surname; date of birth; and postcode. This data set of CTAS records constituted the primary cases affected by the event, described henceforth as the 'delay group'. A control group consisting of all primary cases from the same time window (30 September to 5 October 2020 inclusive) that were unaffected by the event was used as a comparison. The event affected exclusively cases arising from community testing sites and not hospital or other testing sites; therefore, primary cases in the control group were restricted to those arising from community testing.[9]

### Identifying cases and contacts

Contact records were linked to the case who reported them and assigned to the same group (delay or control group) as their associated case. Secondary cases were defined as (1) individuals reported as a contact by a primary case and (2) having a contact event with the primary case between 2 and 14 days inclusive prior to the onset of symptoms of the secondary case (or test date if no symptom onset available). If the secondary case was a household contact of the primary case, the date of interaction was taken to be the date of symptom onset (or test date if no symptom onset available) of the primary case. Secondary cases may be exposed to more than one primary case; therefore, only one transmission event was chosen to link a secondary case to the primary case who most likely infected them based on the following hierarchy: (1) a household transmission event was prioritised above all other types of contact event (due to the higher risk of transmission in this setting); (2) where multiple events of the same priority occurred, the most recent exposure was selected as the transmission event. Individuals who had taken multiple SARS-CoV-2 tests generated a CTAS record for each positive test result. These were identified by matching on name, date of birth, NHS number and contact details. Only the record with the earliest test date was retained, as contact tracing should have been undertaken on receipt of the first positive result.

### Linkage to health outcomes

Contact tracing records were linked to mortality data provided by UKHSA which describe death in people with a positive SARS-CoV-2 test. Matching was performed using NHS number. Two definitions of mortality consistent with SARS-CoV-2 surveillance were used: (1) death of a person with a laboratory-confirmed positive SARS-CoV-2 test and whose death was within 60 days of the first specimen date or more than 60 days after the first specimen date if COVID-19 is mentioned on the death certificate and (2) death of a person with a laboratory-confirmed positive

SARS-CoV-2 test and whose death was within 28 days of the first positive specimen date.[10]

Contact data sets were also linked to the UKHSA hospital-onset COVID-19 data set (extracted on 22 November 2021), which pulls from identifiable daily feeds of data on hospitalisations: Secondary Uses Services (a national data set describing patient hospitalisations); Emergency Care Data Set (a national data set describing patient use of urgent and emergency care); and COVID-19 testing data. The data set provides one hospital spell per case if they had a hospital admission or discharge in the 14-day period before or after their first positive SARS-CoV-2 test, including people already in hospital at the time of test. Matching was performed using NHS number, date of birth, name and sex.

### Descriptive analysis

Analysis was performed in R V.1.3.1056. Investigators LF, LP, CT, AW and CC had access to the study data. Demographic information and contact tracing outcomes were described for the case and contact data sets. Time taken for contact tracing was estimated using different dates in the contact tracing process: dates of test, laboratory report, SGSS record creation, CTAS record creation and contact tracing completion (online supplemental figure 2). Time taken to initiate the contact tracing process was defined as the days between the laboratory report date of the case and the date of upload of the case or contact record into CTAS. Time taken to complete contact tracing was defined as the days between the test date of the case and the date of contact tracing completion of the case or contact. Contact tracing completion was defined as the status of the individual's CTAS record being set to 'complete' by a call handler or the individual submitting an online contact tracing form.

### Regression analysis

To understand the effect of the event on transmission, adjusted secondary attack rates (SARs) and adjusted ORs of infection with SARS-CoV-2 among primary and secondary contacts were estimated via postestimation from a logistic regression model. The predictor under investigation was whether the exposing case was in the delay or the control group. After review of crude SARs stratified by contact setting (household or non-household), effect modification of the predictor (delay or control group) by the contact setting (household or non-household) was allowed for in the model. The potential confounders included in the model were age group, sex, geographic region (one of nine UKHSA-defined geographical areas) of the contact and whether the contact completed contact tracing.

To understand the effect of the event on health outcomes, ORs for experiencing hospitalisation or mortality among primary and secondary contacts were calculated using logistic regression with Firth penalisation, comparing individuals in the delay and control groups. When individuals had contact episodes with multiple cases, the earliest episode per individual was retained. A very small number of mortality events was reported; therefore, unadjusted, crude ORs are presented to avoid overfitting the regression model, and the covariate balance between the delay and control groups suggests any major confounding is unlikely. As a sensitivity analysis, health outcomes of contacts in delay group were compared with a subgroup of contacts in the control group who were contact traced rapidly, defined as within 3 days of the test date of the associated case.

### Patient and public involvement

Patients or the public were not involved in the design, or conduct, or reporting, or dissemination plans of this research.

## RESULTS
### Sample characteristics

Overall, 15 861 SGSS records were identified as affected by the event and 15 467 (98%) were matched to a CTAS record (online supplemental table 1). Following data cleaning, 15 285 (96%) primary cases affected by the event remained eligible for inclusion in the study (online supplemental figure 3). A control group of 43 742 concurrent primary cases was created, consisting of all CTAS records from the time period of 30 September to 5 October 2020 inclusive which were not affected by the event. Secondary and tertiary cases and primary and secondary contacts were identified in the CTAS database. Demographic information is described in online supplemental tables 2 and 3.

### Nature of contact tracing delay

For primary cases in the delay group, it took on average 2 days longer for their records to be uploaded to CTAS, initiating contact tracing, after the laboratory report date of their positive test result (table 1, figure 1). Primary cases took on average 3 days longer to complete contact tracing in the delay group (6 days, IQR: 4–7) than the control group (3 days, IQR: 2–5) after their date of positive test. For primary contacts in the delay group, it took on average 3 days longer for their records to be uploaded to CTAS, relative to the laboratory report date of their associated case. It took 3 days longer to complete contact tracing for primary contacts in the delay group (6 days, IQR: 5–8) than the control group (3 days, IQR: 2–5), relative to the test date of their associated case (table 2, figure 1). Once records were uploaded onto CTAS, the primary cases and contacts in the delay group took the same median amount of time to complete contact tracing as the control group (1 day).

The proportion of cases for whom contact tracing was completed was slightly lower for primary cases in the delay group (80%, 95% CI: 79% to 81%) than the control group (83%, 95% CI: 83% to 84%), a difference of −3.5% (95% CI: −4.2% to −2.8%) (table 1). There was no difference in the proportion of secondary and tertiary

**Table 1** Contact tracing outcomes for primary, secondary and tertiary cases

| Cases | Delay group | | | Control group | | |
|---|---|---|---|---|---|---|
| | Primary | Secondary | Tertiary | Primary | Secondary | Tertiary |
| Observations (n) | 15285 | 2748 | 382 | 43742 | 9575 | 1335 |
| Individuals (n (%)) | 15285 (100.0%) | 2695 (98.1%) | 375 (98.2%) | 43742 (100.0%) | 9382 (98.0%) | 1307 (97.9%) |
| Completed contact tracing (n (%, 95% CI)) | | | | | | |
| Yes | 12221 (80.0%, 79.3% to 81%) | 2137 (77.8%, 76.2% to 79.3%) | 276 (72.3%, 67.5% to 76.7%) | 36503 (83.5%, 83.1% to 83.8%) | 7291 (76.1%, 75.3% to 77.0%) | 971 (72.7%, 70.3% to 75.1%) |
| Difference in proportions who completed contact tracing (delay group minus control group) (95% CI) | 3.5% (−4.2% to −2.8%) | 1.6% (−0.2% to 3.4%) | 0.5% (−5.7% to 4.8%) | N/A | N/A | N/A |
| Median (and IQR) number of contacts reported per case | | | | | | |
| All | 3 (1–4) | 3 (1–4) | 3 (1–5) | 3 (1–4) | 3 (1–4) | 3 (1–4) |
| Household | 2 (1–3) | 2 (1–3) | 3 (1–4) | 2 (1–3) | 2 (1–4) | 2 (1–3) |
| Non-household | 0 (0–1) | 0 (0–0) | 0 (0–0) | 0 (0–1) | 0 (0–0) | 0 (0–0) |
| Mean number of contacts reported per case | | | | | | |
| All | 3.0 | 3.0 | 3.6 | 3.2 | 3.1 | 3.2 |
| Household | 2.2 | 2.4 | 2.6 | 2.3 | 2.5 | 2.4 |
| Non-household | 0.8 | 0.6 | 1.0 | 0.9 | 0.6 | 0.7 |
| Number of cases with ≥2 household contacts (n (%, 95% CI)) | 7361 (48.2%, 47.4% to 49.0%) | 1348 (49.1%, 47.2% to 50.9%) | 182 (47.6%, 42.5% to 52.8%) | 22244 (50.9%, 50.4% to 51.3%) | 4717 (49.3%, 48.3% to 50.3%) | 631 (47.3%, 44.6% to 50.0%) |
| Median (and IQR) time taken for contact tracing, days between | | | | | | |
| SGSS lab report date to CTAS upload date | 3 (2–4) | 1 (0–2) | 1 (0–1) | 1 (0–1) | 1 (0–1) | 1 (0–1) |
| Test date to CTAS completion date | 6 (4–7) | 4 (2–5) | 4 (2–5) | 3 (2–5) | 3 (2–5) | 3 (2–5) |
| Mean time taken for contact tracing, time between | | | | | | |
| SGSS lab report date to CTAS upload date | 3.4 | 1.3 | 1.1 | 1.0 | 0.8 | 0.8 |
| Test date to CTAS completion date | 6.1 | 4.1 | 4.2 | 3.5 | 3.7 | 3.7 |

CTAS, Contact Tracing and Advisory Service; SGSS, Second Generation Surveillance System.

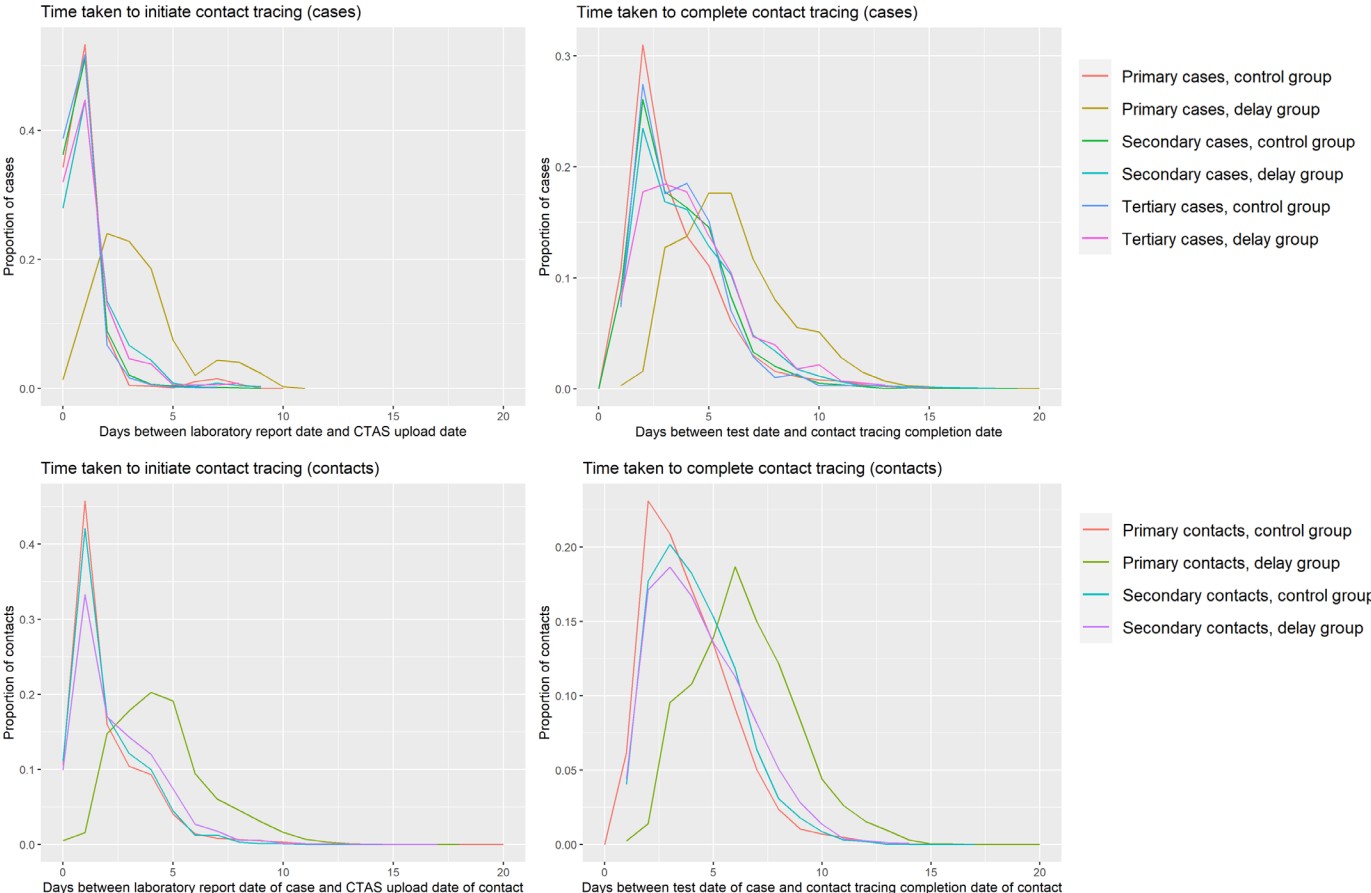

**Figure 1** Graphs describing the time taken to initiate and complete contact tracing of cases and contacts in the delay and control groups. CTAS, Contact Tracing and Advisory Service.

cases who completed contact tracing. The proportion of primary contacts traced was lower by 6.5% (95% CI: −7.1% to −5.9%) in the delay group than the control group (delay group 51%, 95% CI: 51% to 52%; control group 58%, 95% CI: 57% to 58%). There was no evidence of a difference for secondary contacts (delay group 51%, 95% CI: 50% to 53%; control group 52%, 95% CI: 51% to 52%) (table 2). For primary and secondary cases, the median numbers of household and non-household contacts per case were the same in the delay and control groups; for tertiary cases, the median number was the same for non-household contacts but was greater by one person for household contacts in the delay compared with the control group (table 1). For contacts, the nature of contact was reported as household contact for 71%–79% of contacts in each group (table 2).

### Effect on transmission

There was no evidence of a difference between the delay and control groups in the overall adjusted SARs among primary or secondary contacts (table 3). However, differences were observed when stratified by household or non-household setting; SARs in non-household secondary contacts were higher for the delay group (7.9%, 95% CI: 6.5% to 9.2%) than the control group (5.9%, 95% CI: 5.3% to 6.6%), corresponding to 1.36 (95% CI: 1.09 to 1.69) times greater odds of infection. No difference

was observed between the groups among household secondary contacts. For primary contacts, household SARs were lower for the delay group than for the control group. Conversely, for non-household contacts, the SARs in primary contacts were higher for the delay group (6.3%, 95% CI: 5.8% to 6.8%) than for the control group (5.7%, 95% CI: 5.5% to 5.9%), corresponding to 1.12 (95% CI: 1.02 to 1.22) greater odds of infection.

### Effect on health outcomes

COVID-19 death was a rare outcome in both groups with no evidence of a difference in 28-day or 60-day mortality rates among primary or secondary contacts in the delay and control groups (table 4). Using the 28-day definition of mortality, for primary contacts, there were 2.9 deaths (95% CI: 1.4 to 5.4) per 10 000 in the delay group compared with 3.7 deaths (95% CI: 2.7 to 5.1) per 10 000 in the control group. For secondary contacts, this was 0.0 (95% CI: 0.0 to 6.1) deaths per 10 000 in the delay group compared with 1.4 (95% CI: 0.3 to 4.1) deaths per 10 000 in the control group. There was no evidence for a difference in the odds of mortality for the delay group compared with the control group (table 4, figure 2).

The proportion of contacts admitted to hospital as inpatients within 14 days of their first SARS-CoV-2 test was almost identical in the delay and control groups, for primary contacts (delay 3.3% (95% CI: 3.1% to 3.5%),

**Table 2** Contact tracing outcomes for primary and secondary contacts

| Contacts | Delay group | | Control group | |
|---|---|---|---|---|
| | Primary | Secondary | Primary | Secondary |
| Observations (n) | 37 373 | 6540 | 121 205 | 23 119 |
| Individuals, n (%) | 34 047 (91.1%) | 6045 (92.4%) | 109 907 (90.7%) | 21 201 (91.7%) |
| Completed contact tracing (n (%, 95% CI)) | | | | |
| Yes | 19 072 (51.0%, 50.5% to 51.5%) | 3361 (51.4%, 50.2% to 52.6%) | 69 702 (57.5%, 57.2% to 57.8%) | 11 915 (51.5%, 50.9% to 52.2%) |
| Difference in proportions who completed contact tracing (delay group minus control group) (95% CI)) | 6.5% (−7.1% to −5.9%) | 0.1% (−1.5% to 1.2%) | N/A | N/A |
| Nature of contact (n (%, 95% CI)) | | | | |
| Household | 27 477 (73.5%, 73.1% to 74.0%) | 5152 (78.8%, 77.8% to 79.8%) | 85 538 (70.6%, 70.3% to 70.8%) | 18 221 (78.8%, 78.3% to 79.3%) |
| Non-household | 9635 (25.8%, 25.3% to 26.2%) | 1371 (21.0%, 20.0% to 22.0%) | 34 699 (28.6%, 28.4% to 28.9%) | 4609 (19.9%, 19.4% to 20.5%) |
| NA | 261 (0.7%) | 17 (0.3%) | 968 (0.8%) | 289 (1.3%) |
| Median (and IQR) time taken for contact tracing, days between | | | | |
| SGSS lab report date of associated case and contact's upload date | 4 (3–6) | 2 (1–4) | 1 (1–3) | 1 (1–3) |
| Test date of associated case and contact's completion date | 6 (5–8) | 4 (3–6) | 3 (2–5) | 4 (3–5) |
| Mean time taken for contact tracing, days between | | | | |
| SGSS lab report date of associated case and contact's upload date | 4.5 | 2.4 | 2.0 | 2.0 |
| Test date of associated case and contact's completion date | 6.4 | 4.4 | 3.9 | 4.2 |

SGSS, Second Generation Surveillance System.

**Table 3** Adjusted secondary attack rates (SARs) and ORs of infection among contacts of primary and secondary cases in the delay group compared with the control group, by setting of contact

| Setting of contact event | Delay | | | Control | | | Adjusted* OR of infection (95% CI)† |
|---|---|---|---|---|---|---|---|
| | Contacts (n) | Contacts becoming cases | Adjusted* SAR (95% CI) | Contacts (n) | Contacts becoming cases | Adjusted* SAR (95% CI) | |
| **Primary contacts** | | | | | | | |
| All | 37 373 | 2702 | 7.5% (7.3% to 7.8%) | 121 205 | 9419 | 7.7% (7.5% to 7.8%) | 0.99 (0.94 to 1.03) |
| Household | 27 477 | 2063 | 8.1% (7.8% to 8.4%) | 85 538 | 7195 | 8.6% (8.4% to 8.7%) | 0.94 (0.89 to 0.99) |
| Non-household | 9896 | 639 | 6.3% (5.8% to 6.8%) | 35 667 | 2224 | 5.7% (5.5% to 5.9%) | 1.12 (1.02 to 1.22) |
| **Secondary contacts** | | | | | | | |
| All | 6540 | 377 | 5.8% (5.2% to 6.3%) | 23 119 | 1315 | 5.7% (5.4% to 6.0%) | 0.99 (0.88 to 1.12) |
| Household | 5152 | 256 | 5.1% (4.5% to 5.7%) | 18 221 | 990 | 5.6% (5.3% to 6.0%) | 0.91 (0.79 to 1.05) |
| Non-household | 1388 | 121 | 7.9% (6.5% to 9.2%) | 4898 | 325 | 5.9% (5.3% to 6.6%) | 1.36 (1.09 to 1.69) |

*Adjusted SARs and ORs allowing for effect modification of the study type by household, and adjusted for the age group, sex and geographic region of the contact and whether they completed contact tracing.
†Adjusted OR of infection, comparing the odds of SARS-CoV-2 infection among contacts in the control group (reference group) to the delay group.

**Table 4** ORs for mortality and hospital outcomes in the delay group compared with the control group

| Primary contacts | Delay group | | Control group | | Crude OR* (95% CI) |
|---|---|---|---|---|---|
| | Number (n=34 047) | Proportion (95% CI) | Number (n=109 907) | Proportion (95% CI) | |
| Deaths 28-day definition | 10 | 2.9 (1.4 to 5.4) per 10000 population | 41 | 3.7 (2.7 to 5.1) per 10000 population | 0.8 (0.4 to 1.6) |
| Deaths 60-day definition | 10 | 2.9 (1.4 to 5.4) per 10000 population | 61 | 5.6 (4.2 to 7.1) per 10000 population | 0.6 (0.3 to 1.1) |
| Admission to hospital as inpatient | 1121 | 3.3% (3.1% to 3.5%) | 3364 | 3.1% (3.0% to 3.2%) | 1.1 (1.0 to 1.2) |
| **Secondary contacts** | Number (n=21 201) | Proportion (95% CI) | Number (n=6045) | Proportion (95% CI) | Crude OR* (95% CI) |
| Deaths 28-day definition | 0 | 0.0 (0.0 to 6.1) per 10000 population | 3 | 1.4 (0.3 to 4.1) per 10000 population | 0.5 (0.0 to 9.7) |
| Deaths 60-day definition | 1 | 1.7 (0.0 to 9.2) per 10000 population | 7 | 3.3 (1.3 to 6.8) per 10000 population | 0.7 (0.1 to 4.0) |
| Admission to hospital as inpatient | 206 | 3.4% (3.0% to 3.9%) | 687 | 3.2% (3.0% to 3.5%) | 1.1 (0.9 to 1.2) |

*The control group is the reference group (OR=1).

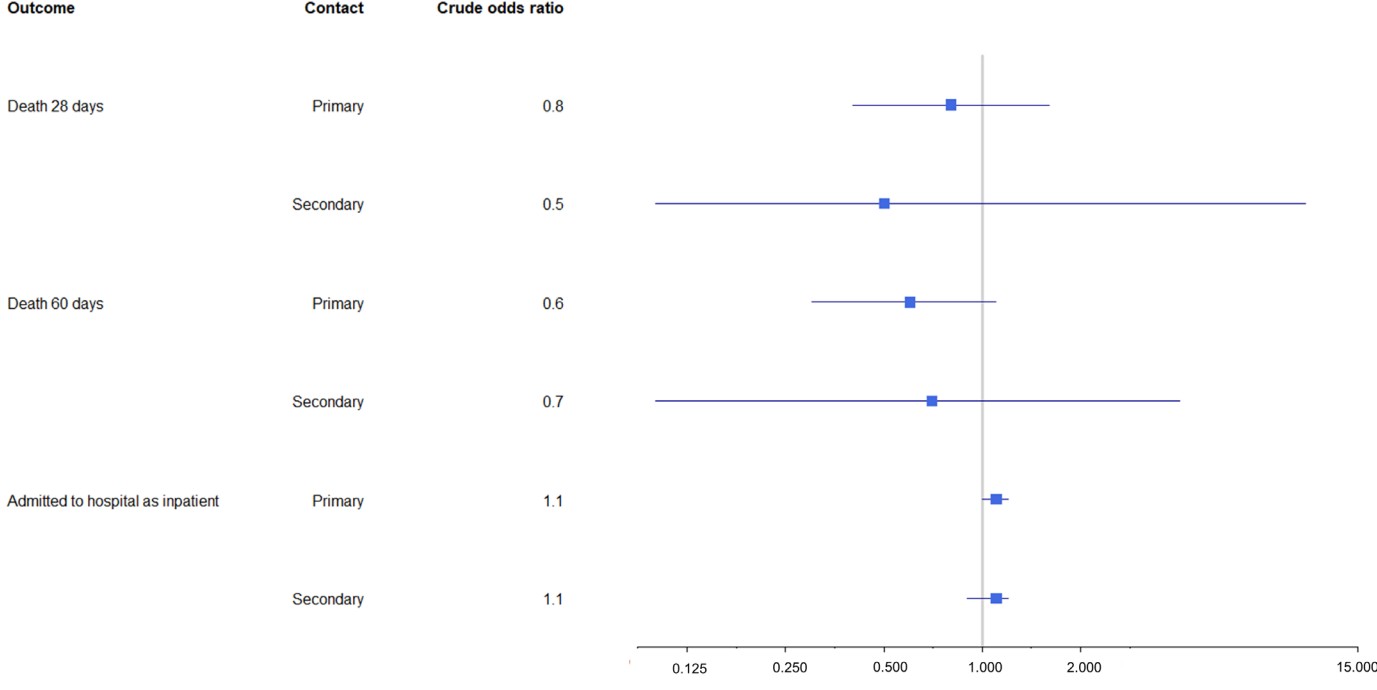

**Figure 2** Graph describing ORs for mortality and hospitalisation, comparing the delay group to the control group (reference OR=1).

control 3.1% (95% CI: 3.0% to 3.2%)) and secondary contacts (delay 3.4% (95% CI: 3.0% to 3.9%), control 3.2% (95% CI: 3.0% to 3.5%)) (table 4). There was no significant difference in the crude or adjusted odds of hospital admission between the delay and control groups (table 4, figure 2). Additionally, there was no significant difference in the crude or adjusted odds of hospitalisation or death when the delay group was compared with a subset of the control group who were contact traced most rapidly (online supplemental table 4).

## DISCUSSION
### Statement of principal findings
The contact tracing of 15861 SARS-CoV-2 cases affected by the event was delayed by an average of 3 days. There was evidence that the delay was associated with a slight decrease in the proportion of cases and contacts who completed contact tracing. Despite the delay, the mean and median number of overall contacts and non-household contacts made per case were similar in the delay and control groups, for primary, secondary and tertiary cases, and the median number of household contacts per case was near-identical.

As expected, we observed a higher SAR from secondary to tertiary cases, for non-household contact events in the delay group. There was no evidence of a difference in the SARs or odds of infection for secondary to tertiary cases overall or for household contact events. We found no evidence of a difference in hospitalisation or mortality among primary or secondary contacts in the delay or control group using the standard definitions for COVID-19-related hospitalisation and mortality used for surveillance purposes in the UK.

### Strengths and weaknesses
The event acted as a natural experiment whereby one group of individuals by chance experienced delayed contact tracing, enabling a comparison of their outcomes with concurrent, unaffected cases and contacts to assess the impact of contact tracing more generally. Having a single national system collating all test results and a single national contact tracing system facilitated the study. Through record linkage, we could identify successive generations of contacts and cases and describe key health outcomes associated with the delay in contact tracing; however, some outcomes were rare and it is possible that we did not have the power to detect a small difference in hospitalisation or mortality.

Secondary transmission could only be estimated using contacts reported by cases who met the contact definition.[11] These would not include unknown contacts; however, the similarity in the number of reported contacts between the two groups suggests that unknown contacts are also likely to be similar. Contact tracing was not completed for a minority of cases in the delay and control groups, so there are likely to be further transmission events that are unknown and not described, and subsequent under-reporting of contacts. People who do not engage in contact tracing differ from those who do in terms of ethnicity and socioeconomic status; however, this is unlikely to differ between delay and control groups.[12–14] Completion of contact tracing was slightly lower in the delay group; for primary contacts, this was

51% in the delay group and 58% in the control group. This could have potentially increased the likelihood of worse outcomes in this group.

It is important to note that there was an inherent lag in the initiation of contact tracing for all cases; therefore, the delay group was not compared with a perfect example of contact tracing, which may explain the lack of difference in observed health outcomes. To assess this, we conducted sensitivity analyses comparing the delay group to a subset of the control group who initiated contact tracing most quickly, but we still did not detect a difference in health outcomes and results were consistent with the main analysis. This could be partly because secondary and tertiary contacts in the delay group did not appear to experience a significant delay in initiating or completing tracing compared with the control group. Additionally, some cases had multiple CTAS records, for example, if they tested several times; we retained only the earliest episode. It is possible that some cases in the delay group reported subsequent positive tests soon after their first test which were not affected by the event (eg, after being admitted to hospital). Therefore, these people might still have been contact traced in a timely manner.

Finally, there is likely to be high uncertainty when tracing the complex network of contacts from the community transmission stage of the pandemic. The actual risk of infection among contacts is likely to vary significantly depending on degree of contact, which was not described. Some transmissions were likely yielded by superspreading events with a disproportionate impact on the pandemic, which were not captured in our comparison of SARs. Notably, primary cases were defined as being either affected by the event (delay group) or from the same time period (control group). Because the event occurred over a period of a small number of days, during which time transmission events occurred, primary cases generated some secondary and tertiary cases which were also classed as primary cases themselves and may have been directly affected by the event. Therefore, transmission chains may have been affected at different stages. Additionally, with regard to household transmission chains, it is worth noting that because many secondary cases were exposed to the virus in their household, their onward household attack rates could be lowered as their exposer would not become a case again within the time period studied.

## Strengths and weaknesses in relation to other studies

A previous modelling exercise using aggregate published data projected substantial adverse health outcomes, describing a drastic rise in SARS-CoV-2 infections and deaths in areas most affected by the event and estimating over 125 000 additional infections and 1500 additional COVID-19-related deaths.[6] Their rapid modelling approach described broader regional trends in infections and deaths using publicly available population surveillance data from the nearest calendar week to the event. This has the advantage of capturing all individuals possibly affected by the event, including any unknown or unreported contacts. However, it could lead to an overestimation of the health impact of the event. Our study described outcomes of the cases known to be affected by the event, which were a subset of the cases from 25 September to 2 October 2020, but the previous modelling study defined affected cases as all those with specimen dates from 20 to 27 September 2020; this might also lead to less accurate estimation of health impacts. It is also important to note that the event occurred at a point in the pandemic when incidence of COVID-19 was increasing nationally and more rapidly in some regions.[15 16]

## Meaning of the study

Contacts of SARS-CoV-2 cases are at increased risk of infection, and this risk is higher among household contacts.[17–19] It is also known that significant numbers of infected people are asymptomatic and can transmit the virus.[17 18 20] Therefore, tracing all contacts of cases of SARS-CoV-2 to inform them of their exposure and give them advice on measures to reduce the risk of onward transmission has been a cornerstone of the pandemic response. This event offered an opportunity to assess the impact of contact tracing on health outcomes. Interestingly, few differences in health outcomes were observed between people impacted by the delay and concurrent cases and contacts.

There was no difference in the number of contact events. This may reflect that people have generally maintained low levels of contacts throughout the pandemic.[20] We did observe an increase in secondary transmission among non-household contacts, which is to be expected if people are not aware of their exposure and therefore not isolating. However, non-household contacts comprised a minority of reported contacts. Household contacts made up the majority of transmission events and did not experience greater secondary transmission, perhaps due to the more limited effect of self-isolation in preventing transmission within households. This could explain in part why we did not see such a large effect on transmission and health outcomes as predicted by other research.

## Unanswered questions and future research

Contact tracing is an effective tool to control transmission of infection.[21–23] Modelling studies have estimated the benefit from contact tracing and self-isolation to reduce SARS-CoV-2 transmission.[24–26] There is more limited evidence on the actual impact on transmission of the very large-scale contact tracing undertaken during the pandemic. For contact tracing to be effective, it needs to be timely and reach, as far as possible, all contacts.[27–29] The lack of a significant adverse health impact observed in our study could reflect the fact that there was an inherent delay in the tracing of all cases, which has been shown to reduce the effectiveness of SARS-CoV-2 contact tracing.[30 31] The lack of a difference may also reflect that, at that time, people generally were limiting their contact with others, with low numbers of non-household contacts

made by individuals in both groups.[20] Other factors that may have contributed include cases informing their contacts directly of their exposure status and need for self-isolation prior to official notification from the contact tracing system, and that a majority of contact events occurred in the household, where self-isolation is less likely to prevent transmission. All of these factors suggest that large-scale contact tracing may have had a limited impact on SARS-CoV-2 transmission at that time. Future research should evaluate the most effective approaches to conducting large-scale contact tracing.

## CONCLUSIONS

We did not find a significant impact on population health outcomes in individuals affected by a delay in contact tracing.

**Author affiliations**
[1]UK Health Security Agency South of England, Bristol, UK
[2]Population Health Sciences, University of Bristol, Bristol, UK
[3]UK Health Security Agency East of England, Cambridge, UK
[4]UK Health Security Agency North of England, Leeds, UK
[5]MRC Biostatistics Unit, Cambridge, UK
[6]UK Health Security Agency, London, UK
[7]UK Health Security Agency Midlands and East of England, Birmingham, UK

**Acknowledgements**  We acknowledge the UK Health Security Agency (UKHSA) for permitting and facilitating access to data. We acknowledge Anne-Marie O'Connell, Richard Dunn, Tom Clare, Fernando Capelastegui, Russell Hope, Simon Collin and Efejiro Ashano (UKHSA) and James Thomas (NHS Digital) for assisting with data acquisition and linkage. We acknowledge Daniela De Angelis (MRC Biostatistics Unit, UKHSA) for contributing to project development and feedback. We acknowledge Oliver McManus (UKHSA) for assisting with code review.

**Contributors**  LF, LP and IO contributed to conception and design of the work, analysis and interpretation of data and drafted the manuscript. CT, AW, CC, CA and GH contributed to design of the work, and acquisition, analysis and interpretation of data. SS, PS, MH, OE and AC contributed to design of the work and interpretation of data. LF is the guarantor. All authors revised the manuscript critically and approved the final version of the manuscript.

**Funding**  The authors have not declared a specific grant for this research from any funding agency in the public, commercial or not-for-profit sectors. LF, LP, AC, MH and IO acknowledge support from the NIHR Health Protection Research Unit in Behavioural Science and Evaluation at University of Bristol. PS was funded by the NIHR Programme Grants for Applied Research programme (grant RP-PG-0616-20008). SS was funded by UKRI (grant MC_UU_00002/10) and UKHSA. For the purpose of open access, the author has applied a Creative Commons Attribution (CC BY) licence to any Author Accepted Manuscript version arising.

**Disclaimer**  The views expressed are those of the author and not necessarily those of the NIHR, the Department of Health and Social Care, or the UKHSA.

**Competing interests**  None declared.

**Patient and public involvement**  Patients and/or the public were not involved in the design, or conduct, or reporting, or dissemination plans of this research.

**Patient consent for publication**  Not applicable.

**Ethics approval**  Approval for the study was granted from the Public Health England (PHE) (now UKHSA) Research Ethics and Governance Group (REGG). R&D reference: R&D 431. Permission was obtained from UKHSA to access contact tracing, hospitalisation and mortality data. Individual consent was not required as the study presented anonymised, population-level data.

**Provenance and peer review**  Not commissioned; externally peer reviewed.

**Data availability statement**  Data are available upon reasonable request. The data that support this study were collected as part of a public health response and are considered sensitive and not made publicly available. Reasonable requests for access to anonymised data and data dictionary will be considered by the authors on request.

**ORCID iDs**
Lucy Findlater http://orcid.org/0000-0003-1736-7661
Cong Chen http://orcid.org/0000-0003-0355-3930
Shaun Seaman http://orcid.org/0000-0003-3726-5937
Pantelis Samartsidis http://orcid.org/0000-0002-4491-9655
Andre Charlett http://orcid.org/0000-0001-7154-0432
Gareth Hughes http://orcid.org/0000-0002-3781-0117
Matthew Hickman http://orcid.org/0000-0001-9864-459X
Obaghe Edeghere http://orcid.org/0000-0002-4275-6338
Isabel Oliver http://orcid.org/0000-0002-6106-1734

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
