## [Reviewer comments · BMJ Open]

ARTICLE DETAILS

TITLE (PROVISIONAL)	Evaluating the impact on health outcomes of an event that resulted in a delay in contact tracing of COVID-19 cases in England, September 2020: an observational study
AUTHORS	Findlater, Lucy; Pierotti, Livia; Turner, Charlie; Wensley, Adrian; Chen, Cong; Seaman, Shaun; Samartsidis, Pantelis; Charlett, Andre; Anderson, Charlotte; Hughes, Gareth; Hickman, Matthew; Edeghere, Obaghe; Oliver, Isabel

VERSION 1 – REVIEW

REVIEWER	Rodrigo Quiroga Universidad Nacional de Córdoba, Química Teórica y Computacional
REVIEW RETURNED	07-Jul-2022

GENERAL COMMENTS	Evaluating the impact on health outcomes of an event that resulted in a delay in contact tracing of COVID-19 cases General comments ===== I enjoyed reviewing this interesting paper, and I commend the authors for this work. It is particularly interesting to be able to contrast the previously published modelling study with real world data. However, I am recommending that a minor review is necessary before acceptance, due to some issues that I will highlight individually below. --Specific comments-- Major comments ----- 1- Within the limitations I was expecting to read some things that were not present in the provided list, and which may all result in a diminished difference in CT effectiveness observed between delay and control groups. I would urge the authors to consider including some of the following limitations: a) Small difference in the proportion of primary contacts traced (51% in delay group versus 58% in control group). b) Tracing of secondary and tertiary contacts for the delay group was executed timely c) Delayed contact tracing could have no differential effect if the individuals from the delay group personally contacted their secondary contacts to inform them of their positive test and advised isolation d) Delayed contact tracing would not be expected to produce a differential effect on household contacts, since positive test results were informed normally
---

e) The amount of non-household contacts reported per case is very small. This may be due to a very strict definition of non-household close contacts, and this could limit the effectiveness of contact tracing as possibly infected individuals were never treated as close contacts.

f) The mean delay between test date of an associated case and contact's completion date of 4 days for the control group is a large amount which could already be diminishing contact tracing effectiveness (see 1- [https://www.thelancet.com/journals/lanpub/article/PIIS2468-2667\(20\)30157-2/fulltext](https://www.thelancet.com/journals/lanpub/article/PIIS2468-2667(20)30157-2/fulltext), 2- <https://www.nature.com/articles/s41598-021-93538-5>, and 3- <https://journals.plos.org/ploscompbiol/article?id=10.1371/journal.pcbi.1009122>), which could also therefore diminish any differences expected to be observed between delay and control groups.

2- While I agree that the data shown supports the conclusion that "The delay in contact tracing had a limited impact on population health outcomes" I do not agree with the reasoning behind said conclusion. The discussion states that "We observed an increase in secondary transmission among non-household contacts but this was small". In my opinion, the difference in SARs between delay and control group was not small. I would say it is actually a rather large difference (7.9 versus 5.9 indicates an OR of 1.34). This is especially true when taking into account all the possible confounding factors and limitations mentioned above. Also, this is almost exactly the estimated increase in infections estimated by Fetzer et al in the modelling study. I infer that the data shows that the main reason Fetzer et al estimate a much larger impact due to the delay than what actually occurred, is the fact that they estimated a 33% increase for all close contacts instead of only non-household secondary contacts, which actually conform a minority of total close contacts. Therefore, data indicates the main reason that "The delay in contact tracing had a limited impact on population health outcomes" is that a significant increase in SAR was only observed for non-household contacts, which only comprised 21% of the total secondary close contacts.

3- Regarding the measured impact on hospitalization and mortality, why would the authors expect differences between delay and control groups? The increase in deaths predicted by Fetzer et al is mainly due to increased infections due to delayed contact tracing, and not due to increased chance of mortality or severe outcomes in the direct contacts of the delay group. Also, even if one expected to find differences, the analysis needs to account for a wide range of factors that are known to have a profound impact on mortality, with the main one being age.

Minor comments

1- I in the Results section the authors claim that "There was no evidence of a difference between the delay and control groups in the overall adjusted secondary attack rates (SARs) in primary or secondary contacts". Why do the authors think that household SARs should differ between delay and control groups? I think the most relevant information here is actually regarding the non-household contacts, and for that group an increase of 34% in SAR for the delay group is actually very significant.

2- In Table 1, I think an odds ratio column with its respective 95% CI would be much more informative than a "Difference in attack rates"

	column.
--	---------

REVIEWER	Chi-Tai Fang National Taiwan University
REVIEW RETURNED	27-Aug-2022

GENERAL COMMENTS	The authors reported data on secondary transmission from contacts of primary cases after an unexpected delay in contact tracing. These data are of interest. However, the statistically negative results should be interpreted very carefully. Although the authors already acknowledge important limitations in interpreting these results in discussion. I would like to remind several key considerations not well addressed in current version of manuscript: (1) High uncertainty in tracing the extremely complex contact network during generalized community transmission stage of the pandemic (misclassification of secondary and tertiary cases and contacts and substantial underreporting of contacts); (2) High variations in actual risk among contacts (which made sample size and power calculation inherently difficult); (3) High variations in impact of secondary transmission (some secondary transmissions yielded superspreading events with an out-of-proportion impact on the epidemic, in another word, NOT all secondary transmissions are equal, therefore, a simple comparison of secondary attack rates could have missed the point). The authors may want to incorporate the above comment into their already excellent discussion on the limitation of this work.
--

VERSION 1 – AUTHOR RESPONSE

Reviewer: 1

Dr. Rodrigo Quiroga, Universidad Nacional de Córdoba Comments to the Author:

Evaluating the impact on health outcomes of an event that resulted in a delay in contact tracing of COVID-19 cases

General comments

=====

I enjoyed reviewing this interesting paper, and I commend the authors for this work. It is particularly interesting to be able to contrast the previously published modelling study with real world data. However, I am recommending that a minor review is necessary before acceptance, due to some issues that I will highlight individually below.

Thank you for your review.

--Specific comments--

Major comments

1- Within the limitations I was expecting to read some things that were not present in the provided list, and which may all result in a diminished difference in CT effectiveness observed between delay and control groups. I would urge the authors to consider including some of the following limitations:

- a) Small difference in the proportion of primary contacts traced (51% in delay group versus 58% in control group).
 - We have stressed this more clearly in the Discussion: Strengths and weaknesses, paragraph 2: "Completion of contact tracing was slightly lower in the delay group; for primary contacts, 51% in the delay group and 58% in the control group completed the contact tracing process."
- b) Tracing of secondary and tertiary contacts for the delay group was executed timely
 - We have stressed this more clearly in the Discussion: Strengths and weaknesses, paragraph 3: "To assess this, we conducted sensitivity analyses comparing the delay group to a subset of the control group who initiated contact tracing more quickly than the average for the control group, but we still did not detect a difference in health outcomes and the results were consistent with the main analysis. This could be because secondary and tertiary contacts in

the delay group did not appear to experience a significant delay in initiating or completing tracing compared to the control group.”

- c) Delayed contact tracing could have no differential effect if the individuals from the delay group personally contacted their secondary contacts to inform them of their positive test and advised isolation
- We have now stressed this more clearly in the Discussion, Unanswered questions and future research: “Other factors that may have contributed include cases informing their contacts directly of their exposure status and need for self-isolation prior to official notification from the contact tracing system.”
- d) Delayed contact tracing would not be expected to produce a differential effect on household contacts, since positive test results were informed normally
- We have stressed that the delay is less likely to have caused an effect in household contacts because self-isolation is less effective.
 - We could also suggest that cases are more likely to inform their household contacts of their positive test result and need for isolation, but this is an assumption that might not be the case, for example, in larger shared houses. We think the below sentence in the Discussion captures the ideas of informing cases directly and household contacts being less affected by a delay.
 - Discussion, Unanswered questions and future research: “Other factors that may have contributed include cases informing their contacts directly of their exposure status and need for self-isolation prior to official notification from the contact tracing system, and that a majority of contact events occurred in the household, where self-isolation is less likely to prevent transmission.”
- e) The amount of non-household contacts reported per case is very small. This may be due to a very strict definition of non-household close contacts, and this could limit the effectiveness of contact tracing as possibly infected individuals were never treated as close contacts.
- We think that the low numbers of non-household contacts made are because people were generally limiting their contact with others at that stage of the pandemic (Sep 2020), which is supported by other research at that time.
(UKHSA 2021 - Weekly statistics for NHS Test and Trace (England): 14 October to 20 October 2021.
https://assets.publishing.service.gov.uk/government/uploads/system/uploads/attachment_data/file/1029536/test-and-trace-oct-14-to-20.pdf)
 - We have stressed this a bit more clearly in the Discussion, Unanswered questions and future research: “The lack of a difference may also reflect that, at the time, people generally were limiting their contact with others, with low numbers of non-household contacts made by individuals in both groups”
- f) The mean delay between test date of an associated case and contact’s completion date of 4 days for the control group is a large amount which could already be diminishing contact tracing effectiveness (see 1-
[4](https://eur01.safelinks.protection.outlook.com/?url=https%3A%2F%2Fjournals.plos.org%2Fploscompbiol%2Farticle%3Fid%3D10.1371%2Fjournal.pcbi.1009122&data=05%7C01%7CLucy.Findlater%40ukhsa.gov.uk%7Ccbe3781a78c94f191dc608da9278a2ff%7Cee4e14994a354b2ead475f3cf9de8666%7C0%7C0%7C637983344071196315%7CUnknown%7CTWFpbGZsb3d8eyJWljoic4wLjAwMDAiLCJQIjoiV2luMzliLCJBTiI6Ikl1haWwiLCJXVCi6Mn0%3D%7C3000%7C%7C%7C&sd=nf8RokMMJIQbWcsJtTCuerkYqGiprLLGedmkvtcxQw%3D&reserved=0), which could also therefore diminish any differences expected to be observed between delay and control groups.<div data-bbox=)

- We have mentioned briefly that cases and contacts experience an inherent lag in the contact tracing process, but we have now inserted the references provided and stressed this more clearly.
- Discussion, Unanswered questions and future research: “The lack of a significant health impact observed in our study could reflect the fact that there was an inherent delay in the tracing of all cases, which has been shown to reduce the effectiveness of COVID-19 contact tracing (references)”

2-While I agree that the data shown supports the conclusion that “The delay in contact tracing had a limited impact on population health outcomes” I do not agree with the reasoning behind said conclusion.

The discussion states that “We observed an increase in secondary transmission among non-household contacts but this was small”. In my opinion, the difference in SARs between delay and control group was not small. I would say it is actually a rather large difference (7.9 versus 5.9 indicates an OR of 1.34). This is especially true when taking into account all the possible confounding factors and limitations mentioned above. Also, this is almost exactly the estimated increase in infections estimated by Fetzer et al in the modelling study. I infer that the data shows that the main reason Fetzer et al estimate a much larger impact due to the delay than what actually occurred, is the fact that they estimated a 33% increase for all close contacts instead of only non-household secondary contacts, which actually conform a minority of total close contacts.

Therefore, data indicates the main reason that “The delay in contact tracing had a limited impact on population health outcomes” is that a significant increase in SAR was only observed for non-household contacts, which only comprised 21% of the total secondary close contacts.

- This is an interesting conclusion, thank you. We have now specified clearly that there was a notable difference in SARs for non-household contacts, but these comprised a minority of the total transmission events; most transmission occurred in households which did not appear to be as impacted by the delay. This could explain (in part) why we did not see such a large effect on transmission as predicted by Fetzer et al.
- Discussion, Meaning of the study: “We did observe an increase in secondary transmission among non-household contacts, which is to be expected if people are not aware of their exposure and therefore not isolating. However, non-household contacts comprised a minority of reported contacts. Household contacts made up the majority of transmission events and did not experience greater secondary transmission, perhaps due to the more limited effect of self-isolation in preventing transmission within households. This could explain in part why we did not see such a large effect on transmission and health outcomes as predicted by other research.”

3- Regarding the measured impact on hospitalization and mortality, why would the authors expect differences between delay and control groups? The increase in deaths predicted by Fetzer et al is mainly due to increased infections due to delayed contact tracing, and not due to increased chance of mortality or severe outcomes in the direct contacts of the delay group. Also, even if one expected to find differences, the analysis needs to account for a wide range of factors that are known to have a profound impact on mortality, with the main one being age.

- We expected that there could be an increase in hospitalisation and mortality amongst primary and secondary contacts, because if more of those people became infected, more of those people could be hospitalised and die. Because our denominator here is primary/secondary contacts, and not secondary/tertiary cases, we are hoping to capture increased chance of infection and chance of hospitalisation/death amongst contacts, rather than the chance of severe outcomes amongst cases.
- It is also possible that there could have been a delay in seeking help amongst primary/secondary contacts who did not know they were exposed and possibly infected in the usual timeframe. (Introduction: “In addition, there could potentially be delays in secondary cases seeking healthcare support, increasing their risk of hospitalisation or excess mortality.”)
- We did not end up adjusting for confounding factors, such as age, when describing mortality because of the reason given in the Methods, Regression analysis paragraph: “A very small number of mortality events was reported; therefore, unadjusted, crude odds ratios are presented to avoid overfitting the regression model, and the covariate balance between the delay and control groups suggests any major confounding is unlikely. As a sensitivity analysis, the health outcomes of the contacts in delay group were compared to a subgroup of contacts in the control group who were contact traced rapidly, defined as within three days of the date of test of the associated case.”

- We did calculate adjusted odds ratios for mortality initially, but we were advised by our co-author Andre Charlett (deputy director of statistics, modelling and economics at UKHSA) to provide crude odds ratios only due to low numbers of the reported outcome and risk of overfitting the model. We did not see a notable difference in odds ratios for mortality when calculating adjusted odds ratios when compared to the crude odds ratios.

Minor comments

1- I in the Results section the authors claim that “There was no evidence of a difference between the delay and control groups in the overall adjusted secondary attack rates (SARs) in primary or secondary contacts”. Why do the authors think that household SARs should differ between delay and control groups? I think the most relevant information here is actually regarding the non-household contacts, and for that group an increase of 34% in SAR for the delay group is actually very significant.

- We also thought that there would likely be a difference in non-household SARs for the delay and control groups, but not a difference in household SARs because household transmission is less affected by self-isolation. However, it still seems helpful to provide SARs stratified by household and non-household transmission to support our hypothesis that there wasn't a notable difference in household transmission risk between the delay and control group.
- With regard to the increased of 34% in SAR for non-household contacts in the delay group, we were hesitant to over-state the significance of this result given that the confidence intervals overlap for the estimates for the delay and control groups. This increase in transmission is mentioned in the Statement of principal findings in the Discussion: “As expected, we observed a higher SAR from secondary to tertiary cases, for non-household contact events in the delay group”.
- It is also mentioned in the Results section of the Abstract: “The SAR for non-household contacts was higher amongst secondary contacts in the delay group than the control group (delay group: 7.9%, 95%CI:6.5% to 9.2%; control group: 5.9%, 95%CI: 5.3% to 6.6%)”

2- In Table 1, I think an odds ratio column with its respective 95% CI would be much more informative than a “Difference in attack rates” column.

- We have replaced the “difference in attack rates” column with an “adjusted odds ratio of infection” column and we have changed the methods and results text accordingly. The odds ratio describes the difference in the odds of infection with SARS-CoV-2 between the delay and control groups, adjusted for various confounders as described in the text.
- Methods, Regression analysis: “To understand the effect of the event on transmission, adjusted secondary attack rates (SARs) and adjusted odds ratios of infection with SARS-CoV-2 amongst primary and secondary contacts were estimated via post-estimation from a logistic regression model.”
- Results, Effect on transmission: “SARs in non-household secondary contacts were higher for the delay group (7.9%, 95% CI 6.5% to 9.2%) than the control group (5.9%, 95% CI 5.3% to 6.6%), corresponding to 1.36 (1.09 to 1.69) times greater odds of infection. No difference was observed between the groups amongst household secondary contacts. For primary contacts, household SARs were lower for the delay group than for the control group. Conversely, for non-household contacts, the SARs in primary contacts were higher for the delay group (6.3%, 95% CI 5.8% to 6.8%) than for the control group (5.7%, 95% CI 5.5% to 5.9%), corresponding to 1.12 (1.02 to 1.22) greater odds of infection.”
- Results, Table 3: “Table 3: Adjusted secondary attack rates and odds ratios of infection amongst contacts of primary and secondary cases in the delay group compared to the control group, by setting of contact”. Table, final column: “Adjusted* odds ratio of infection (95% CI)**” (with updated numbers). Caption: “*Adjusted SARs and odds ratios allowing for effect modification of the study type by household, and adjusted for the age group, sex and geographic region of the contact, and whether they completed contact tracing. **Adjusted odds ratio of infection, comparing the odds of SARS-CoV-2 infection amongst contacts in the control group (reference group) to the delay group”

Reviewer: 2

Prof. Chi-Tai Fang, National Taiwan University Comments to the Author:

The authors reported data on secondary transmission from contacts of primary cases after an unexpected delay in contact tracing. These data are of interest. However, the statistically negative

results should be interpreted very carefully. Although the authors already acknowledge important limitations in interpreting these results in discussion.

Thank you for your review.

I would like to remind several key considerations not well addressed in current version of manuscript:

(1) High uncertainty in tracing the extremely complex contact network during generalized community transmission stage of the pandemic (misclassification of secondary and tertiary cases and contacts and substantial underreporting of contacts);

- We have now added this point to the Discussion, Strengths and weaknesses: “Finally, there is likely to be high uncertainty when tracing the complex network of contacts from the community transmission stage of the pandemic”
- And in Strengths and weaknesses: “Contact tracing was not completed for a minority of cases in the delay and control groups, so there are likely to be further transmission events that are unknown and not described, and subsequent underreporting of contacts”

(2) High variations in actual risk among contacts (which made sample size and power calculation inherently difficult);

- We have now revised the Discussion, Strengths and weaknesses: “Additionally, the actual risk of infection amongst contacts is likely to vary significantly depending on degree of contact, which is not captured in this study.”

(3) High variations in impact of secondary transmission (some secondary transmissions yielded superspreading events with an out-of-proportion impact on the epidemic, in another word, NOT all secondary transmissions are equal, therefore, a simple comparison of secondary attack rates could have missed the point). The authors may want to incorporate the above comment into their already excellent discussion on the limitation of this work.

- We have now revised the Discussion, Strengths and weaknesses: “Some transmissions were likely yielded by superspreading events with a disproportionate impact on the pandemic, which were not captured in our comparison of secondary attack rates”